



# Response of cirrus clouds to idealised perturbations from aviation

Ella Gilbert[1,2], Jhaswantsing Purseed[3], Yun Li[4], Martina Krämer[5,6], Beatrice Altamura[3] & Nicolas Bellouin[1,3]

[1] Department of Meteorology, University of Reading, United Kingdom
[2] Now at: British Antarctic Survey, Cambridge, United Kingdom
[3] Institut Pierre Simon Laplace / CNRS, Sorbonne Université, Paris, France
[4] Institute of Energy and Climate Research: Troposphere, Research Centre Jülich, Jülich, Germany
[5] Johannes-Gutenberg University Mainz, Institute for Atmospheric Physics of the Atmosphere, Mainz, Germany
[6] Institute of Energy and Climate Research: Stratosphere, Research Centre Jülich, Jülich, Germany

*Correspondence to*: Ella Gilbert (ellgil82@bas.ac.uk)

**Abstract.** Aviation is a rapidly growing source of climate forcing, and the non-$CO_2$ effective radiative forcing of aviation is approximately twice that of aviation $CO_2$. However, considerable uncertainty remains regarding aviation's non-$CO_2$ effects because the radiative forcing of aviation aerosol-cloud interactions, especially with cirrus clouds, is poorly known. Here, we
use a large eddy simulation model to quantify the impact of ice crystal number concentration (ICNC) perturbations on the water budget and microphysics of pre-existing cirrus clouds. These perturbations aim to represent the second half of the chain of effects linking aircraft aerosol emissions to changes in ICNC and ice water path. We examine two types of cirrus: warm conveyor belt outflow and gravity wave cirrus, which represent different updraft regimes and formation mechanisms. In both cases, the primary effect of an idealised increase in ICNC is to extend cloud lifetime, with the increase proportional
to the magnitude of the ICNC perturbation applied. The effect is more pronounced in the gravity wave cirrus case than in the warm conveyor belt outflow cirrus case because the latter has lower initial ICNC and ice water contents. Quantitatively, the sensitivity of ice water path (IWP) to changes in ICNC, expressed as $\Delta\ln(IWP)/\Delta\ln(ICNC)$, is 0.06 for gravity wave cirrus and 0.35 for warm conveyor belt outflow cirrus when calculated 45 minutes after imposing the ICNC perturbation. These results suggest that aviation has the potential to increase the lifetime and radiative effects of pre-existing cirrus clouds.

## 1 Introduction

Despite an initial drop during and after the COVID pandemic, global air traffic volumes continue to increase, meaning that if the world reduces emissions in line with climate targets, the difficult-to-decarbonise aviation sector will begin to account for a larger proportion of total radiative forcing (RF) (Lee et al., 2021). Aviation currently accounts for 2 – 3% of global anthropogenic fossil carbon dioxide ($CO_2$) emissions, producing a global mean net effective RF of 34.3 mW m$^{-2}$ (with a 90%
confidence interval of 28 to 40 mW m$^{-2}$) for 2018 (Lee et al., 2021). However, aviation's non-$CO_2$ effective RF of 66.6 (21 –



111) mW m$^{-2}$ is approximately twice as large as the $CO_2$ effective RF. Non-$CO_2$ effects include the effects of emissions of nitrogen oxides (NOx), water vapour, soot and sulphate aerosols, and increased cloudiness due to contrail formation (Lee et al., 2021). However, Lee et al. (2021) was unable to provide a best estimate and uncertainty range for the effective RF of aviation aerosol-cloud interactions because of the lack of consensus in the literature. Yet, according to climate models, the RF estimated to come from aerosol-cloud interactions, both with low-level and high-level clouds, could be as large as several 100s of mW m$^{-2}$, therefore having the potential to dwarf the RF from $CO_2$ and the other non-$CO_2$ effects, including contrails (see section 4.6 and Figure 5 in Lee et al., 2021). Note that aviation aerosol-cloud interactions are distinct from the formation of contrail cirrus, whose RF is counted separately.

The reasons behind the lack of consensus on aviation aerosol-cloud interaction RF comes from large disagreements among climate model estimates. Uncertainties around aviation-aerosol-cirrus interactions arise mainly due to difficulties in simulating background rates of heterogeneous and homogeneous ice nucleation in large-scale models, differences in the treatment of cirrus dynamics – especially updraft velocities – and importantly, the nucleation efficiency of aircraft soot (Lee et al., 2021).

Firstly, the competition between heterogeneous and homogeneous ice nucleation impacts the formation, evolution and properties of contrail cirrus clouds (Spichtinger & Gierens, 2009b; Unterstrasser & Gierens, 2010) and therefore their radiative effects. Recent studies have determined that both heterogeneous and homogeneous ice nucleation occur within cirrus clouds, although for some time there were contrasting ideas about the dominant mechanism (for example, compare Czizco et al., 2013 and Kärcher & Lohmann et al., 2002 or Sölch & Kärcher, 2011). It is now apparent that different ice nucleation mechanisms dominate in different cirrus temperature, aerosol and meteorological regimes (Fan et al., 2016; Krämer et al., 2016; 2020; Froyd et al., 2022) and can inhibit one another by competing for available water vapour (Spichtinger & Gierens 2009a; 2009b; Penner et al., 2018). In regions dominated by heterogeneous freezing and a low aerosol background, model configurations where aviation aerosols increase ice crystal number concentrations (ICNC) globally simulate a strongly positive RF associated with aerosol-cloud interactions. Conversely, in regions dominated by homogeneous freezing, model configurations where aviation aerosols lead to reductions in ICNC simulate negative RFs, which get very strong (more than –300 mW m$^{-2}$) when atmospheric soot background is low (Penner et al., 2018; Righi et al., 2021).

A second source of uncertainty in aerosol-cirrus interactions is the treatment of cirrus dynamics, especially updrafts (Lee et al., 2021). Updraft velocities influence atmospheric cooling rates, and so impact heterogeneous and homogeneous ice nucleation rates (Barahona et al., 2017). Understanding and simulating updrafts in models is therefore crucial for representing cloud properties like ICNC and radiative effects (Penner et al., 2018). Parameterisations of updraft velocity are especially important in the kind of large-scale models that are typically used to make global estimates of the radiative effect



of aerosol-cirrus interactions (e.g., Penner et al., 2018; Gettelman & Chen, 2013) because the coarse resolution of these models means they cannot represent the fine-scale vertical motions that produce fine-scale spatial variability in cloud properties.

Thirdly, the efficiency of aged soot aerosol to act as INPs (ice nucleating particles, which freeze heterogeneously) hours to
days after emission is uncertain (Kärcher et al., 2021; Righi et al., 2021). Studies that assume soot is an efficient INP show that aircraft modify cirrus ICNC and coverage (Urbanek et al., 2018; Zhou & Penner, 2014), whereas those assuming soot is an inefficient INP show much smaller effects (Gettelman & Chen, 2013; Kärcher et al., 2021). Constraining the degree to which soot particles induce the nucleation of cloud ice in the atmosphere is a research priority, and the subject of ongoing research. Kanji et al. (2020) show that soot aerosol is unimportant for immersion mode freezing, while McGraw et al. (2020)
find that soot aerosol can inhibit homogenous freezing of solution aerosol, thinning cirrus clouds and inducing a cooling effect. Kulkarni et al. (2016) and Gao and Kanji (2022) find that in-cloud or in-contrail processing and aging of soot can alter its ice nucleating abilities at cirrus-relevant temperatures. However, recent measurements that consider a realistic size distribution of aviation soot particles suggest that aviation soot remains a poor INP, even after processing, at cirrus cloud temperatures (Testa et al., 2023).


Adding further complexity, these uncertainties compound one another. For example, assuming a high soot nucleation efficiency, Zhou & Penner (2014) find that their estimates of cirrus radiative effect vary widely (from $-350$ to $+90$ mW m$^{-2}$) as a result of uncertainty related to the concentrations of aerosols and the dominant ice nucleation pathway in the background atmosphere. Righi et al. (2021) find that their model simulations of aviation-soot cirrus RF vary from $-40$ to $+15$ mW m$^{-2}$
depending on the assumed soot ice nucleating efficiency and the supersaturation resulting from updrafts, highlighting the important interactions between dynamics and microphysics. Importantly, Righi et al. (2021) cannot match the large RFs of 100s of mW m$^{-2}$ obtained by Penner et al. (2018) despite sampling a wide range of soot activation fractions and critical supersaturations, indicating that other aspects of cloud modelling have an influence on modelled aviation aerosol-cirrus RF.

Aerosol-cirrus interactions take place in a variety of meteorological conditions, so it is important to characterise these in
different natural cirrus regimes. Krämer et al. (2016; 2020) produced a comprehensive climatology of cirrus clouds in various meteorological settings. They use the climatology to categorise different types of cirrus based on formation mechanism and dynamical situation. In their climatology, they contrast in situ cirrus, where the cloud forms heterogeneously or homogeneously in-place, from liquid origin cirrus, which originate from mixed-phase clouds that freeze as they are lifted. Cirrus can further be classified according to vertical velocity as either slow ($<10$ cm s$^{-1}$) or fast ($>10$ cm s$^{-1}$) updraft cirrus
(Krämer et al., 2020). Examples of fast updraft cirrus include gravity wave clouds and cirriform anvils of convective cloud systems, while slow-updraft liquid origin cirrus include those formed in the warm conveyor belt region of low-pressure systems (Krämer et al., 2020).



Several recent studies examined the impact of aircraft on pre-existing cirrus, mostly in the context of embedded contrails.

For example, Tesche et al. (2016) use satellite cloud retrievals to suggest passing aircraft increase cirrus cloud optical depth. Marjani et al. (2022) developed the method of Tesche et al. (2016) to show that ICNC increases in the areas immediately behind, adjacent to and below an aircraft's flight path by 25 – 54%. They explore the effects only on thin, mainly in situ origin cirrus, defined as those clouds with geometrical thickness smaller than 2 km. The primary effect of aircraft aerosols is to increase ICNC (Kärcher et al., 2021; Verma & Burkhardt, 2022). This is shown indirectly by studies examining the effect

of reducing soot number in aviation fuel and consequently ICNC (e.g., Burkhardt et al., 2018; Voigt et al., 2021). Verma & Burkhardt (2022) use an LES model to show that contrail cirrus formation within existing cirrus can increase ICNC by a few orders of magnitude, especially in optically thick cirrus but also, to a lesser extent, in relatively thin cirrus.

The aerosol-cloud interaction problem can be divided into two steps. The first propagates a change in INPs into a change in

ice crystal number. The second links the change in ice crystal number with changes in cloud properties, and especially ice water content. This study focuses on that second step by using idealised LES modelling to examine the response of cloud microphysical processes, cloud properties and water budget to ICNC perturbations that mimic the addition of INPs that freeze heterogeneously. These perturbations are intended to provide insight into the response of cirrus cloud water budgets, processes and bulk properties to idealised ICNC perturbations. The perturbations induced are intended to be plausible. For

example, we apply ICNC perturbations of 25% and 50%, which are comparable to the results of Marjani et al. (2022).

Details of the modelling framework, case studies and idealised perturbations are given in Sect. 2, while results are presented and discussed in Sect. 3 and conclusions are offered in Sect. 4.

**2 Methods**

**2.1 Modelling Framework**

We use the UK Met Office NERC Cloud model (MONC), which is a LES model based on Bousinnesq-type equations. It includes parameterisation schemes for sub-grid scale turbulence and radiation and is coupled with a multi-moment cloud microphysics scheme called CASIM (Cloud and AeroSol Interacting Microphysics, Field et al., 2023). MONC has periodic

horizontal boundary conditions with a rigid lid at the top and bottom of the domain. Here, the model simulates a 60 x 60 grid point domain, with 100 m horizontal resolution (so a 6 x 6 km domain). Model vertical resolution is approximately 120 m, but model top altitude and the number of levels depends on the cirrus case being simulated, as described below.

Because the focus here is on ice processes and negligible amounts of liquid are expected to be present in these cirrus clouds,

CASIM is configured to simulate double moment ice but only single moment liquid. Heterogeneous ice nucleation is



parameterised following DeMott et al. (2015), which calculates ice mass and number concentrations via contact and immersion freezing mechanisms from prescribed INP profiles using empirically derived relationships. Activation of liquid droplets is switched off, so cloud liquid is initialised by prescribing a fixed initial cloud liquid droplet number of 50 cm$^{-3}$. At the cirrus temperatures simulated (between 220 and 235 K depending on the case), a small fraction of these droplets

immediately freezes homogeneously into ice crystals in the model, with a small additional contribution (much smaller than 1%) from heterogeneous ice nucleation. We emphasise that this method of initialising the model cloud is not necessarily reflective of the actual ice nucleation mechanisms of the two cirrus cases studied.

## 2.2 Case descriptions

We run two cirrus cases representing two different kinds of natural cirrus in the Krämer et al. (2020) climatology discussed in the introduction. The first case is a gravity wave cirrus case, hereafter referred to as GW, which is based on a case study described in Yang et al. (2012). This case is based on a cirrus cloud observed over the contiguous USA in the lee of the Rocky Mountains on 9 March 2000 at approximately 17:30 local time. The synoptic conditions show that weak lifting was present, likely induced by gravity wave disturbance upstream. Within the cloud region simulated by MONC there is a slight

supersaturation with respect to ice of ~10%, which results in the formation of an optically and geometrically thin cirrus of approximately 0.2 and 1 km, respectively, at the start of our simulation. This cloud is fairly warm and low-altitude for a cirrus cloud (mean modelled in-cloud temperature 229 K, formation at 8.5 – 9.5 km altitude). The GW simulations have 191 vertical levels, from the surface to 22.5 km. Wind forcing is applied in one direction only (u wind). We made two simplifications to the MONC configuration of Yang et al. (2012). First, they applied a time-dependent forcing of potential

temperature, but we apply a constant forcing, equal to the average of their time-dependent forcing. Second, we do not include a time-dependent gravity wave forcing. Instead, we initialise with a subsidence profile and then apply a constant water vapour forcing to compensate for depletion of water vapour to ensure that the cloud does not dissipate immediately. The water vapour forcing rate was calculated from an unforced simulation and is applied between altitudes of 6.7 km and 9.7 km, with values of the order $2 \times 10^{-9}$ kg$^{-1}$ kg$^{-1}$ s$^{-1}$.


The second case is a cirrus cloud that forms in the outflow region of a warm conveyor belt associated with a North Atlantic low-pressure system, hereafter referred to as WCB outflow. The WCB outflow case is based on a case observed over northeast Germany at 06:00 UTC on 29 November 2000 described in Spichtinger et al. (2005). The case shows a thick (~3 km) ice supersaturated region and that extremely thin - potentially even sub-visible - cirrus formed within it, with a

geometrical thickness of ~700 m and an IWC of 0.1 – 0.2 mg m$^{-3}$. The WCB outflow simulations have 131 vertical levels, from the surface to 15.9 km. Wind forcing is applied in both directions (u and v), leading to a diagonally moving atmosphere within the simulated domain. To initialise the WCB outflow case, we follow the procedure described in Spichtinger et al. (2005) to derive input profiles of potential temperature, specific humidity and u and v winds from radiosoundings from the

Lindenberg station at 0600 UTC and ERA5 reanalysis (Hersbach et al., 2020). Unlike the GW case, no water vapour forcing
is applied.

All simulations are run for 21600 s (6 hours). The first 1500 s (GW case) and 3500 s (WCB outflow case) are a spin-up
phase and are not included in the analysis of mean properties and quantities. The period of spin-up was determined by
examining time series of various properties, including IWP and ice tendencies (Figure S1). After an initial spike, the IWP,
total ice tendency and ice microphysics tendency settle into a more stable phase. As shown in Figure S1, the total tendency
(ice + snow + graupel total tendency) in both cases rapidly declines over the first 10-100 s, before increasing again to
stabilise at approximately zero by 600 s in the GW case and 1500 s in the WCB outflow case, suggesting that the processes
influencing the cloud water budget are in balance and the cloud has equilibrated into a stable state for its initial
microphysical properties.


### 2.3 Idealised perturbations

To evaluate the response of cirrus water budget, processes and properties to cloud ice perturbations in both cases, we perturb
ICNC and/or ice mass mixing ratio by applying a multiplication factor to their distributions at the end of the spin-up phase.
The multiplication factor is applied instantaneously across the whole model domain and throughout the entire profile, but
because ice exists only in the cirrus layer, this is equivalent to perturbing only the cirrus cloud. A full list of simulations is
given in Table 1.

We apply multiplication factors of 0.1, 0.5, 0.9, 1.1, 1.25, 1.5, 2 and 10 to explore a wide range of possible aviation
perturbations. The ICNC×1.25, ×1.5 experiments are closest to the satellite-based estimates of Marjani et al. (2022), while
the ICNC×0.5 and ×2 experiments represent large but still-plausible perturbations. The ICNC×0.1 and ×10 perturbations are
unrealistic, but useful to study the degree of linearity in ice water response. Table 1 also includes two additional sensitivity
studies that were conducted using the GW case: IWC×2 and ICE×2. These are described in more detail in Sect. 3.4. All
simulations are available from Gilbert, Purseed & Bellouin (2024).







**Table 1**. List of perturbation experiments run for each cirrus type. Perturbations are applied over the whole domain, both horizontally and vertically. All simulations are available at Gilbert, Purseed & Bellouin (2024).

| Parameter perturbed | Gravity Wave cirrus | Warm Conveyor Belt outflow cirrus |
|---|---|---|
| ICNC | ICNC×0.1 | ICNC×0.1 |
| ICNC | ICNC×0.5 | ICNC×0.5 |
| ICNC | ICNC×0.9 | ICNC×0.9 |
| ICNC | ICNC×1.1 | ICNC×1.1 |
| ICNC | ICNC×1.25 | ICNC×1.25 |
| ICNC | ICNC×1.5 | ICNC×1.5 |
| ICNC | ICNC×2 | ICNC×2 |
| ICNC | ICNC×10 | ICNC×10 |
| IWC | IWC×2 | - |
| ICE×2* | ICE×2 | - |

* ICE = both ICNC and IWC


## 3 Results and Discussion

### 3.1 Description and validation of the control cases

#### 3.1.1 Basic cloud properties

The basic properties of the control cirrus cases are shown in Figure 1. The basic GW case is a geometrically thin cirrus cloud
(approximately ~1 km) that forms at 8.5 – 9.5 km altitude. The WCB outflow is of a comparable geometrical thickness, but forms 600 m higher at altitudes of 9.1 – 10.1 km. The WCB case also features lower liquid cloud layers, but we focus only on the outflow region and the upper cirrus layer here.




As shown in Figure 1, cloud properties such as ICNC, IWC and IWP remain approximately constant during 2000 – 8000 s in
the GW case and 4000 - 10000 s in the WCB outflow case. We hereafter refer to this period as the 'stable phase'. During this
stable phase, mean values of ICNC, IWC and IWP are approximately 0.42 cm$^{-3}$, 3.12 mg m$^{-3}$ and 4.0 g m$^{-2}$, respectively, in
the GW case and 0.04 cm$^{-3}$, 0.23 mg m$^{-3}$ and 0.15 g m$^{-2}$, respectively, in the WCB outflow case. In both cases after ~8,000 s,
and even more so after ~10,000 s, the cloud begins to dissipate as ice crystals grow large enough to sediment out of the cloud
and sublimate in the warmer layers below, causing IWP to decline and ice sink tendencies to increase (the green panels in
Figure S1). This period is hereafter referred to as the 'dissipation phase'. The simulated clouds have completely dissipated
by 14000 s for the GW case, while the cloud remains until around 18000 s in the WCB outflow simulation.

Because we initialise the cloud using a prescribed cloud particle number and double moment ice, the cloud forms by
condensation, followed by immediate freezing. This all happens within the first five seconds of the spin-up period.
Homogeneous nucleation dominates in both cases over heterogeneous nucleation because the clouds form at heights where
temperatures are below the homogeneous freezing threshold of –38°C. Mean homogeneous freezing rates in the first minute
of the simulations are 1400 mg kg$^{-1}$ s$^{-1}$ and 220 mg kg$^{-1}$ s$^{-1}$ for the GW and WCB outflow cases, respectively, while mean
heterogenous nucleation rates are virtually negligible at $1.5 \times 10^{-4}$ mg kg$^{-1}$ s$^{-1}$ and $2.8 \times 10^{-4}$ mg kg$^{-1}$ s$^{-1}$, respectively.
Homogenous nucleation rates are relatively higher at the cloud base, whereas heterogenous nucleation is comparatively
higher at cloud top.

Figure 2 shows mean spatial distributions of IWP during the stable phase of the control simulation in both cases. Both cases
exhibit spatial heterogeneities in IWP, but there are differences between the two cases as a result of different wind forcing.
The GW case is initialised with u winds only, which produces the IWP field in Figure 2a with linear bands across the
domain. In contrast, the IWP field in the WCB outflow case exhibits pockets of higher and lower IWP aligned approximately
diagonally in bands parallel to the prescribed wind field at 45°. Such spatial heterogeneity is typical of cirrus clouds and
suggests that the processes determining the characteristics of both cases are realistic. IWP differs considerably between the
two cases: varying between 2.0 and 5.2 g m$^{-2}$ in the GW control (CTRL) simulation (Figure 2a), and between 0.08 and 0.23 g
m$^{-2}$ in WCB outflow CTRL (Figure 2b).





**Figure 1.** Horizontally averaged mean in-cloud conditions for the control simulations for the two cirrus cloud cases simulated in MONC. The gravity wave (GW) case is shown on the left and the warm conveyor belt (WCB) outflow case on the right. Variables shown are (from top to bottom): ice water content, IWC (mg m$^{-3}$), ice crystal number concentration, ICNC (cm$^{-3}$), temperature (K), relative humidity with respect to ice, RH$_{ice}$ (%) and effective radius, r$_e$ (μm). Cloud-free regions are coloured white in all panels except the ones showing RH$_{ice}$.





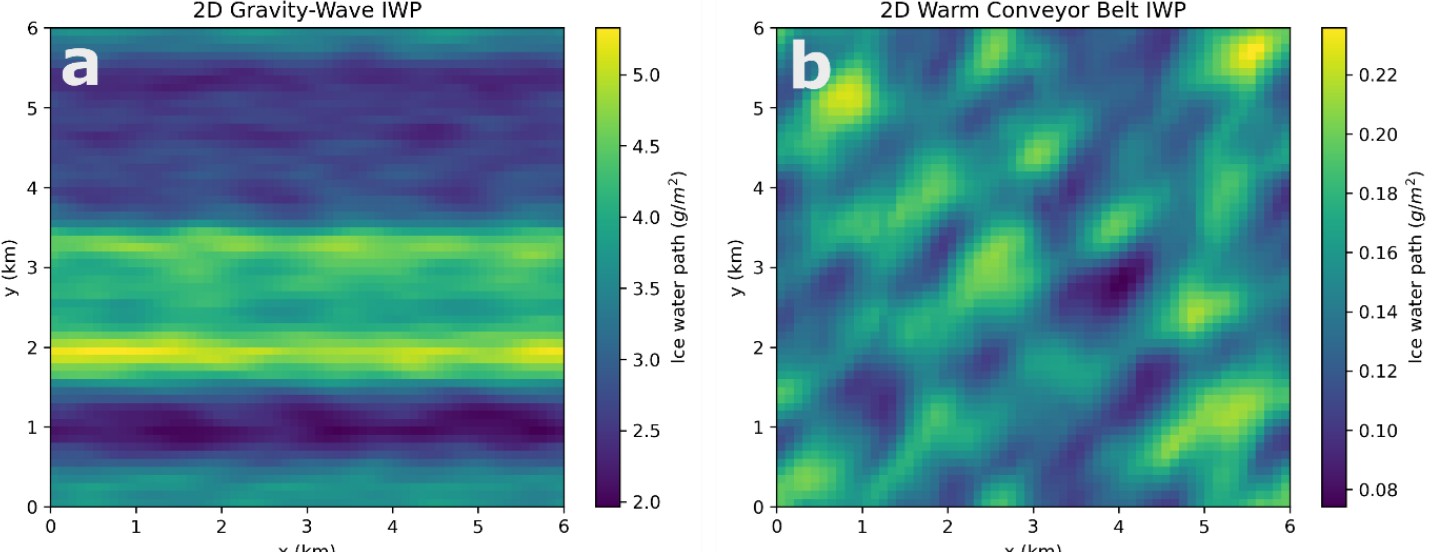

**Figure 2.** Distributions of ice water path (IWP, g m$^{-2}$) averaged over the stable phase of the control simulation for the gravity wave cirrus case (panel a) and the warm conveyor belt outflow case (panel b). Note the different scales for IWP.


### 3.1.2 Comparison with literature

Exact reproduction of the observed and modelled GW (Yang et al., 2012) and WCB outflow (Spichtinger et al., 2005) cases is not expected because of differences between our model configurations and initial conditions. However, Krämer et al. (2016; 2020) and Li et al. (2023) offer an opportunity to compare our simulations to a representative sample of observed

cirrus clouds and hence determine whether the cirrus we simulate can be considered realistic. Table 2 shows summary statistics averaged over the entire domain during the stable phase of all ICNC simulations for both cases, while Figure 3 shows the distribution of ice particle size and number for each control case, with an indication of the size and number ranges that occur most frequently within the cloud. Krämer et al. (2020) show that GW cirrus can be formed in situ or have liquid origin and can be characterised further by their updraft speed.


Taken together, several of the properties of the GW case shown in Table 2 and Figure 3 suggest a high updraft in situ cirrus cloud. According to the classification of Li et al. (2023), which characterises 'mostly liquid origin' and 'mostly in situ origin' cirrus as being above and below the 10 ppmv IWC isoline, respectively, the GW case has properties attributable to both cloud categories. Figure 3a shows that the GW case has high ICNC and relatively small ice particles (the upper left of

the distribution shown in Figure 4 of Krämer et al., 2020). However, the modelled GW cloud is relatively warm and low



altitude compared with other cirrus clouds in this category and has a steady supply of water vapour due to the model configuration. This allows the easy formation of ice crystals and produces high ICNC, despite the comparatively lower simulated median vertical velocity in the GW CTRL (3.34 cm s$^{-1}$) than reported by Li et al. for comparable cirrus types (~10 cm s$^{-1}$). Homogeneous freezing rates dominate in the GW case, and median COT values of 0.08 for the control and 0.11 for the ICNC×2 experiment (Table 2) fit within the typical range of 0.05-1 reported by Krämer et al. (2020) for cirrus in this category. Mean mass-weighted radius ($r_{ice}$) values in the GW case of 12 and 11 µm in the CTRL and ICNC×2 experiments, respectively, are within the 5-15 µm range reported in Krämer et al. (2020). Median IWC and ICNC both fit with the typical fast updraft in situ values in Krämer et al. (2020): IWC ranges between 0.46 mg m$^{-3}$ and 0.81 mg m$^{-3}$ for the control and ICNC×2 experiments, fitting within the range of 0.1 – 7.5 mg m$^{-3}$ shown in Krämer et al. (2020). Meanwhile, GW ICNC of 0.38 and 0.52 cm$^{-3}$, respectively, are consistent with the range of 0.1 – 5 cm$^{-3}$ shown in that study. However, the median ICNC value of 0.38 cm$^{-3}$ in the stable phase of the GW CTRL case is approximately twenty times the median of 0.018 cm$^{-3}$ reported in Li et al. (2023) for "natural cirrus"; and above the 90[th] percentile of 0.17 cm$^{-3}$. Similarly, the GW CTRL median $r_{ice}$ of 12 µm is around three times smaller than the median 42.2 µm shown in Li et al. for the same category, and well below the 10[th] percentile of 24.5 µm.

This combination of relatively high ICNC and small $r_{ice}$ suggests that the GW case occupies the more extreme tails of the distribution of natural cirrus properties. For example, comparing Figure 3a to the observed cloud properties from the ML-CIRRUS campaign shown in Figure 3c in Li et al. (2023), we can see that values typical of the GW case appear in the top left of the distribution. Some of these values fall within the black contour - which encloses 90% of observed ice particles - but many also fall outside this range, with higher ICNC values at smaller ice particle sizes. However, Li et al. (2023) did not observe GW cirrus. Perhaps unsurprisingly, the GW case is not representative of all natural cirrus; rather, it represents a more extreme case.

According to the classification of Krämer et al. (2016), WCB cirrus should be an example of liquid origin cirrus. However, the properties of the WCB case shown in Figure 3b and Table 2 are more typical of a slow updraft in situ cirrus, the category with lowest IWC and ICNC. It is important to note that the WCB case modelled here includes only the outflow region of the cloud and not its lower layers, which impacts its properties. Median IWC of 0.002 and 0.003 mg m$^{-3}$ is in the range of slow updraft in situ cirrus (Krämer et al., 2020), which corresponds to WCB outflow conditions. Similarly, median ICNC of 0.003 and 0.005 for the CTRL and ICNC×2 experiments, respectively, is within the expected range of 0.001-0.02 cm$^{-3}$ for ICNC given in Krämer et al. (2020). R$_{ice}$ values of 15 and 14 µm and COT values of 0.004 and 0.007, respectively, are also more consistent with slow updraft in situ cirrus than liquid origin cirrus. The former is associated with $r_{ice}$ of 15 – 25 µm and COT of 0.001 – 0.05 whereas liquid origin clouds have $r_{ice}$ and COT of 50 – 70 and 1 – 12, respectively. The WCB outflow case, representative of slow updraft background natural cirrus, is one of the most frequently occurring cirrus types in mid-latitudes (Krämer et al., 2020). Considering that these cirrus types are common in mid-latitudes, a region that has a high density of air



traffic, the impacts of increasing ICNC on WCB outflow cirrus may be considered more directly relevant for examining potential aviation-aerosol-cirrus interactions.

It is important to note however, that this direct comparison against the results of Li et al. (2023) is a tough test for the model because observed clouds differ in their meteorological context and stage of evolution, and the sampling strategy is also

different. Aircraft campaigns can involve biased sampling, with mission scientists implicitly sampling thicker clouds and avoiding less favourable cases such as thin cloud or regions of high turbulence or updrafts (Field & Furtado, 2016). The differences between modelled and observed values reported in Li et al. (2023) are therefore difficult to avoid, and the broad agreement between them is encouraging.

**3.2 Cloud evolution**

The simulated cirrus cloud evolves similarly in both cases, developing first into a stable state where sources and sinks of water in the cloud water budget show a small net gain in ice water content (the 'stable phase') before dissipating as a result of sink terms overwhelming source terms (the 'dissipation phase'). The cloud starts dissipating by around 4 and 5.5 hours into the control simulation in the GW and WCB outflow case, respectively. A schematic summarising the magnitude of

processes during both phases is shown in Figure 4.

Later in the lifetime of the cloud, there are far fewer ice crystals and many more snow particles because the crystals have steadily grown: mostly by vapour deposition with some contribution from the accretion of ice onto snow particles. This is also reflected in the auto-conversion term, which represents the transfer of cloud ice particles into snow-sized particles

('large ice crystals' in Figure 4) via diffusion and aggregation and occurs above the threshold of 50 μm (Field et al., 2023). Auto-conversion is one of the largest mean terms in the water budget during the dissipation phase at 1.56 mg kg$^{-1}$ s$^{-1}$ and 0.05 mg kg$^{-1}$ s$^{-1}$ in the GW and WCB outflow cases, respectively, reflecting the growth of particles during this stage. As shown in Figure 4b, sinks of cloud water including sublimation and sedimentation dominate over source terms during the dissipation phase.


This evolution process is also evident in the time series of IWP shown in Figure 5, where IWP in the control simulation remains approximately constant over the first 2-3 hours shown. After this, IWP declines fairly rapidly as the cloud dissipates because ice crystals reach precipitable size and sediment below the cloud base, where they sublimate.

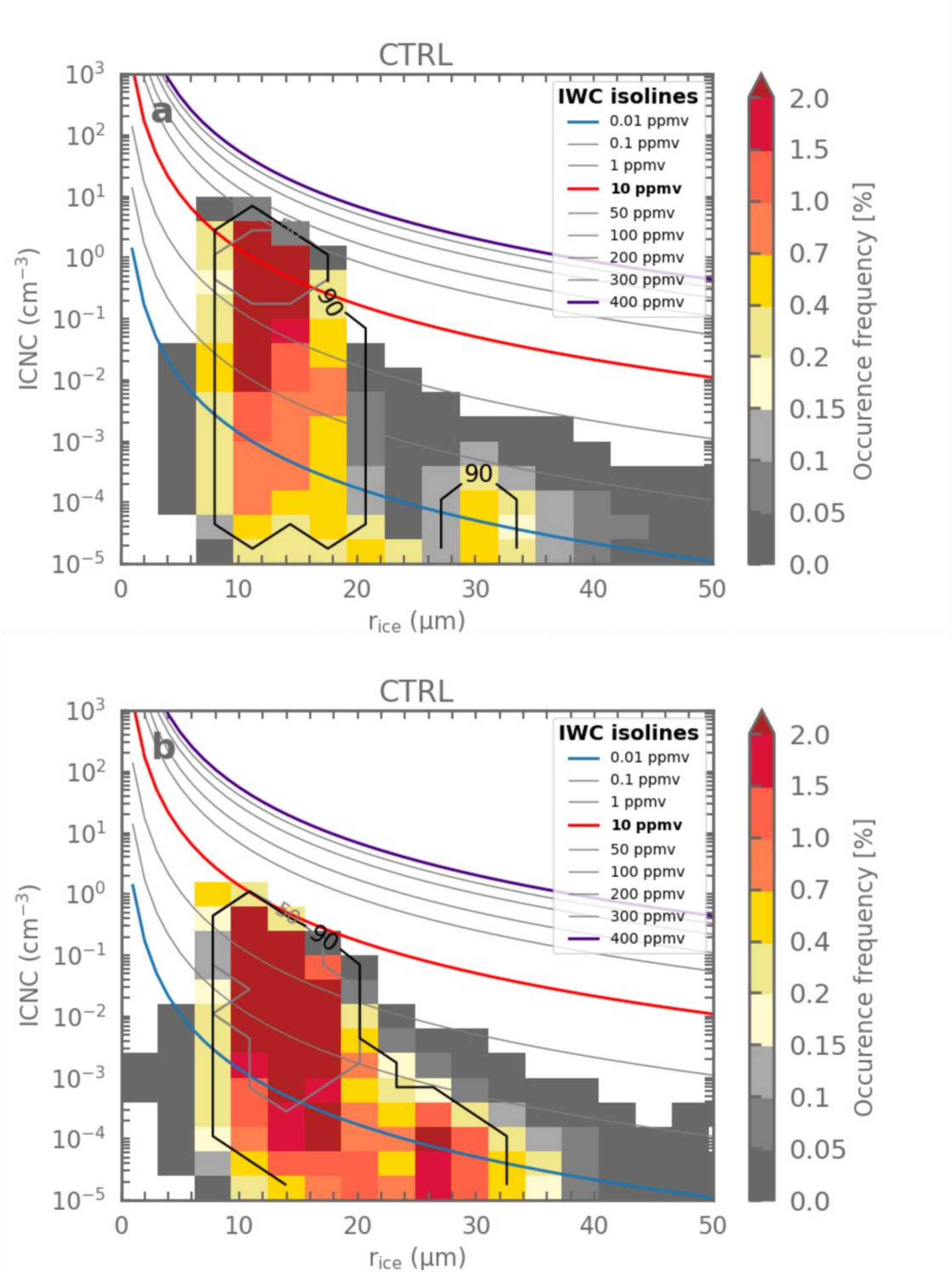

**Figure 3.** Characteristics of the distribution of ice particle numbers and sizes during the stable phases of the Gravity Wave (panel a) and Warm Conveyor Belt outflow (panel b) control cases. Mean mass-weighted radius, $r_{ice}$, is shown on the x axis, while ice crystal number concentration (ICNC) is shown on the y axis. Colours indicate how frequently particles with a given ICNC and $r_{ice}$ are simulated, and the black and grey contours indicate the region where 90% and 50%, respectively, of all ice particles are simulated. Isolines of ice water content (IWC) are also shown as coloured lines.





**Table 2.** Summary statistics for each simulation, expressed as medians over the stable phase. Values for the gravity wave (GW) cirrus simulations are in the top half of the table and those for the warm conveyor belt outflow (WCB) cirrus simulations are in the bottom half. Variables included are temperature (T, in K), updraft speed (w, cm s$^{-1}$), relative humidity over ice (RH$_{ice}$, %), ice water content (IWC, mg m$^{-3}$), ice crystal number concentration (ICNC, cm$^{-3}$), ice crystal effective radius (r$_e$, μm) ice crystal mass mean radius (r$_{ice}$, μm), cloud optical thickness (COT, non-dimensional) and cloud lifetime (s). COT is computed using Equation 1 of Wang et al. (2019) and cloud lifetime is calculated as the first timestep where mean IWC and ICNC have decreased to 5% of their mean over the stable period. All medians are presented as "in-cloud" values, with "in-cloud" defined over grid points with ICNC > 10$^{-5}$ cm$^{-3}$, as per Krämer et al. (2020).

|  |  | T (K) | W (cm s$^{-1}$) | RH$_{ice}$ (%) | IWC (mg m$^{-3}$) | ICNC (cm$^{-3}$) | r$_e$ (μm) | R$_{ice}$ (μm) | COT (nd) | Lifetime (s) |
|---|---|---|---|---|---|---|---|---|---|---|
| | ICNC×0.1 | 228.7 | 2.19 | 71.29 | 0.069 | 0.217 | 25.59 | 10.53 | 0.027 | 2094 |
| | ICNC×0.5 | 228.7 | 2.19 | 71.29 | 0.069 | 0.217 | 25.59 | 10.53 | 0.066 | 9266 |
| | ICNC×0.9 | 228.73 | 3.19 | 71.24 | 0.403 | 0.359 | 30.36 | 12.5 | 0.073 | 11751 |
| | **CTRL** | **228.73** | **3.34** | **71.32** | **0.462** | **0.378** | **30.09** | **12.39** | **0.079** | **12172** |
| GW | ICNC×1.1 | 228.74 | 3.47 | 71.3 | 0.521 | 0.392 | 29.91 | 12.31 | 0.083 | 12549 |
| | ICNC×1.25 | 229.75 | 3.89 | 71.3 | 0.631 | 0.395 | 30.05 | 12.37 | 0.086 | 13094 |
| | ICNC×1.5 | 229.74 | 3.89 | 71.29 | 0.677 | 0.458 | 28.73 | 11.83 | 0.098 | 13730 |
| | ICNC×2 | 228.75 | 4.63 | 71.26 | 0.806 | 0.519 | 27.76 | 11.43 | 0.117 | 15121 |
| | ICNC×10 | 228.75 | 7.62 | 71.42 | 1.370 | 0.958 | 23.97 | 9.87 | 0.247 | 21628 |
| | ICNC×0.1 | 223.72 | 1.138 | 97.38 | 0.0001 | 0.0001 | 66.78 | 27.49 | 0.0009 | 3069 |
| | ICNC×0.5 | 223.72 | 1.138 | 97.32 | 0.0013 | 0.0016 | 36.64 | 15.08 | 0.0023 | 10189 |
| | ICNC×0.9 | 223.72 | 1.152 | 98.32 | 0.0018 | 0.0023 | 36.25 | 14.92 | 0.0034 | 13749 |
| WCB | **CTRL** | **223.72** | **1.15** | **98.31** | **0.0019** | **0.0025** | **35.75** | **14.72** | **0.0037** | **14612** |
| | ICNC×1.1 | 223.72 | 1.149 | 98.3 | 0.0020 | 0.0028 | 35.26 | 14.52 | 0.004 | 15330 |
| | ICNC×1.25 | 223.72 | 1.164 | 98.72 | 0.0021 | 0.0030 | 35.54 | 14.63 | 0.0044 | 16417 |
| | ICNC×1.5 | 223.72 | 1.161 | 98.8 | 0.0023 | 0.0035 | 34.61 | 14.25 | 0.0052 | 18020 |
| | ICNC×2 | 223.72 | 1.157 | 99.09 | 0.0026 | 0.0046 | 34.49 | 14.2 | 0.0067 | 20784 |
| | ICNC×10 | 223.74 | 0.999 | 100.34 | 0.0152 | 0.0419 | 34.57 | 14.23 | 0.0259 | 21645 |




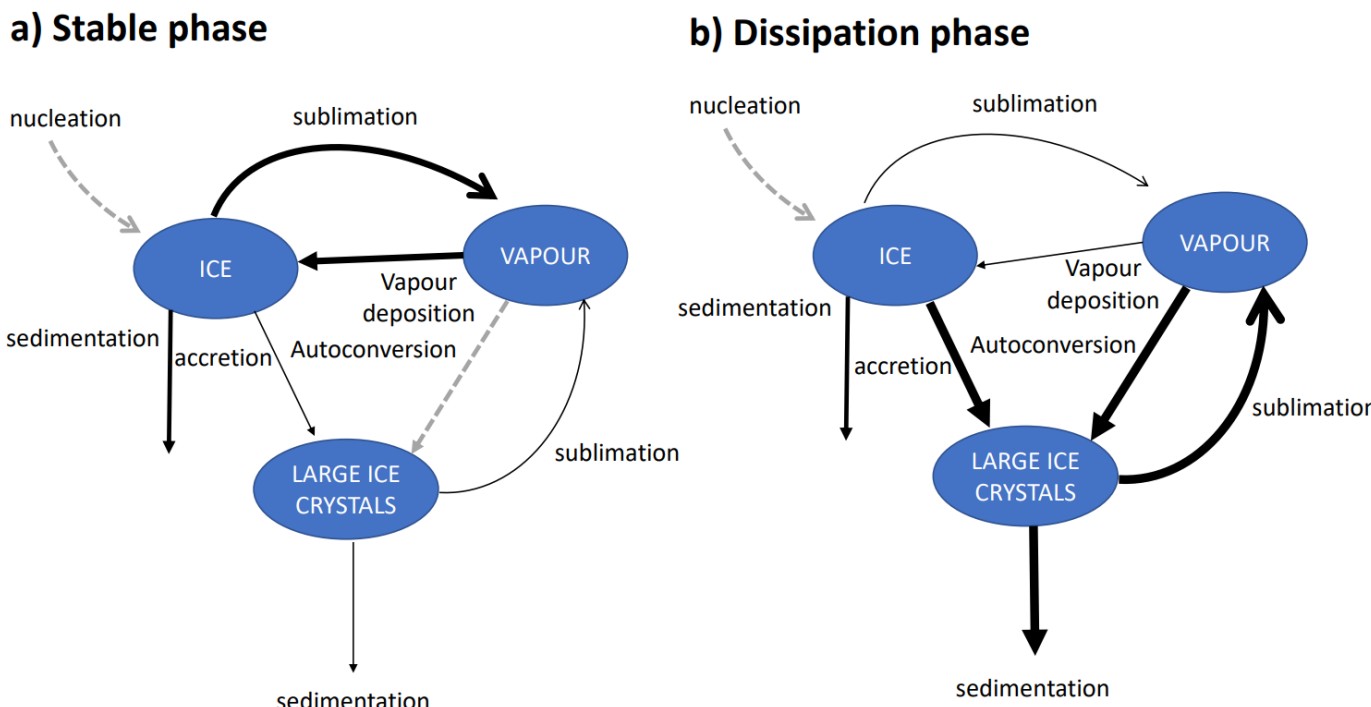

**Figure 4.** Schematic showing a simplified water budget of the cirrus clouds in the control simulation during a) the "stable" phase and b) the "dissipation" phase. The width of the arrows shows the relative magnitude of the mass fluxes in each phase. The "Large ice crystals" category includes snow and graupel particles. The mean values of process rates are given in Table S1.

## 3.3 Impacts of idealised perturbations on bulk cloud properties

Figures 5a and 5b also show time series of IWP in the perturbed GW and WCB outflow cases. The same generic impact of the idealised perturbations is observed in both cases.

When IWC remains fixed, instantaneous perturbations to ICNC cause the same amount of water to be distributed across more crystals, therefore instantly decreasing crystal size when ICNC is increased and, conversely, increasing crystal size when ICNC is decreased. Because there is now a larger number of smaller crystals, the cloud lifetime is increased, as it takes longer for these smaller crystals to grow via vapour deposition and accretion to a size large enough to sediment out of the cloud and sublimate in sub-saturated air. The increased surface area of smaller, more numerous ice crystals results in a higher water vapour deposition flux (see Sect. 3.6 below). Increasing ICNC also increases IWP at a given time relative to the control because of the slower ice crystal loss rates, which results in the longer cloud lifetimes shown in Figure 5a and 5b.





**Figure 5.** Time series of domain-averaged ice water path (IWP, g m$^{-2}$, top row) and in-cloud ice-weighted relative humidity with respect to ice (Ice MMR-weighted RH$_{ice}$, %, bottom row) for the GW case (left column) and WCB outflow case (right column). In panels a and b, the control simulation is shown in black, and the various perturbed simulations are shown with coloured lines as indicated in the legend, while in panels c and d the control is shown as a black dashed line. Note that the GW case uses a water vapour forcing that replenishes water vapour and compensates for its deposition onto ice crystals.





Figure 5c and 5d show time series of relative humidity over ice (RH$_{ice}$) weighted by IWC for the GW and WCB outflow

cases, respectively, so that only in-cloud RH$_{ice}$ is considered. In the GW case (Figure 5c), IWC-weighted RH$_{ice}$ gradually increases over time in all simulations from ~100% at the start of the stable phase (~35 minutes into the simulation) to 100-100.3% by the start of the dissipation phase (~2 hours in CTRL), reaching around 101% by the time the cloud has dissipated. Recall that the GW case uses a water vapour forcing that replenishes water vapour and compensates for its deposition onto ice crystals. IWC-weighted RH$_{ice}$ in the GW perturbation simulations increases more slowly compared to the GW control

when ICNC is increased and increases faster when ICNC is decreased. That is because of the changes in water vapour deposition rates described above which means that IWC grows relatively more slowly. In the WCB outflow control case (Figure 5d), IWC-weighted RH$_{ice}$ increases more rapidly than in the GW case from around 97% at the start of the stable phase to nearly 109% by the time it has completely dissipated. In the perturbed WCB outflow simulations, IWC-weighted RH$_{ice}$ behaves qualitatively like described above for the GW simulations, also in response to changes in water vapour

deposition rates.

In both cases, larger ICNC perturbations have larger impacts. The largest perturbations (ICNC×2 and ICNC×10) increase IWP and cloud lifetime and decrease IWC-weighted RH$_{ice}$ the most, while the smallest positive perturbations (ICNC×1.1) alter these properties the least. Negative perturbations (ICNC×0.9, ICNC×0.5 and ICNC×0.1) have the opposite effect,

shortening cloud lifetime, to such an extent in the ICNC×0.1 case that the clouds dissipate immediately. The effect is consistent across all simulations, suggesting that the change in IWP or IWC-weighted RH$_{ice}$ is proportional to the magnitude of the ICNC perturbation.

As shown in Table 2, the perturbed simulations in both cases have very similar (thermo)dynamical properties compared to

the control, with median temperature, updraft speed and RH$_{ice}$ differences of 0.02 K, 1.29 cm s$^1$ and −0.054 percentage points, respectively for the ICNC×2 GW case and <0.01 K, 0.007 cm s$^{-1}$ and 0.78 percentage points, respectively, for the ICNC×2 WCB outflow case. This suggests that the ICNC perturbations have a minimal effect on thermodynamic and dynamic properties.

Larger differences are simulated in the microphysical properties. For example, in the GW case, the median r$_e$ and r$_{ice}$ in the ICNC×2 experiment is 28 and 11 µm, respectively, compared to 30 and 12 µm in CTRL, a difference of –2.3 and –1.0 µm, respectively. The pattern is the same for the WCB outflow case, with median r$_e$ and r$_{ice}$ of 35 and 14 µm in the ICNC×2 experiment compared with 36 and 15 µm, respectively, in CTRL. As discussed previously, and as shown in Figures 5 and Table 2, the smaller ice crystals make the lifetime of the perturbed cirrus cases longer, particularly for the GW case. Larger

perturbations are associated with longer cloud lifetimes, for example the cloud lifetime is extended by approximately 300 s, 700 s, 1700 s and 3100 s in the ICNC×1.1, ICNC×1.25, ICNC×1.5 and ICNC×2 experiments, respectively, while ICNC×10





extends the lifetime of the cloud beyond the end of the simulation. This effect is also evident in Figure 5 with the WCB outflow case, but in all simulations with positive perturbations greater than ICNC×1.1, the cloud lifetime is longer than the 6-hour simulation period.


## 3.4 Sensitivity simulations

Two sensitivity tests were conducted using the GW case to examine whether IWC perturbations are more efficient than ICNC perturbations at perturbing IWP, and to test the role of ice crystal size. The GW case was used because, as shown in Sect. 3.1, it represents a more extreme case with higher IWC and ICNC than the WCB outflow case. These tests are briefly

described here but not shown in the interests of brevity. In the first of these simulations, IWC was doubled with no change to ICNC (IWC×2), and in the second, ICNC and IWC were both doubled (ICE×2). Both experiments were compared with the GW CTRL and ICNC×2 simulation.

As shown in Sect. 3.3 above, increasing ICNC alone (ICNC×2) increases the lifetime of the cloud because it distributes the

same quantity of water vapour across a larger number of smaller ice crystals. This affects the water budget by slowing down crystal growth, so it takes longer for crystals to reach precipitable size and sediment out of the cloud, and by slowing the rate of loss via sublimation. Increasing IWC alone (IWC×2) causes an initial large spike in IWP and IWC compared to the control simulation, but this extra ice is rapidly lost as the unstable cloud equilibrates to its environment. In the IWC×2 experiment, despite having double the IWC, the cloud lifetime is approximately the same as in CTRL because the larger

crystals simply sublimate and sediment out of the cloud at a faster rate, causing it to dissipate over the same amount of time. Doubling IWC only affects overall cloud lifetime if ICNC is also increased, which is tested in the ICE×2 experiment. ICE×2 has far higher IWP, IWC and ICNC overall, with the cloud lasting longer than the control. However, with this amount of ice, the cloud becomes unstable and does not exhibit a 'stable phase' of approximately constant IWP as shown in Figure S1 for the control cases. IWP declines slowly from 2,000 s to around 12,000 s, after which point it declines rapidly, along

approximately the same gradient as the GW control in its dissipation phase (IWP decreases by 0.83 mg m$^{-2}$ s$^{-1}$ in the control case, compared to 0.94 mg m$^{-2}$ s$^{-1}$ in the ICE×2 experiment, equivalent to approximately 3 g m$^{-2}$ hr$^{-1}$ and 3.3 g m$^{-2}$ hr$^{-1}$, respectively).

Overall, these results suggest that ICNC perturbations have a more pronounced impact on the long-term water budget and

lifetime of the GW cirrus than IWC perturbations, and the results confirm the crucial role in ice crystal size in controlling the response to ICNC perturbations.





**Figure 6.** As in Figure 3, but for the perturbed ICNC×2 simulations of the gravity wave cirrus (a) and warm conveyor belt outflow cirrus (b).



### 3.5 Impact of idealised perturbations on cloud microphysical properties

Figure 6 shows the frequency distribution of ice crystal numbers and sizes in the ICNC×2 simulations for the two cirrus cases, as was shown in Figure 3 for the control simulations. In both cases, the perturbed ice particle distribution has larger numbers of smaller ice particles and the distribution shifts towards lower values of $r_{ice}$ and higher values of ICNC, i.e. towards the top left of the plots in Figures 3 and 6. This is as expected and is consistent with the results shown in Figures 4 and 5 and Table 2.

Specifically, in the stable phase of the ICNC×2 GW simulation, the median ICNC increases to 0.52 cm$^{-3}$ compared to 0.38 cm$^{-3}$ in the control case, while median $r_{ice}$ decreases from 12 µm to 11 µm (Figures 3a and 6a). Similarly, median ICNC increases to $4.6 \times 10^{-3}$ cm$^{-3}$ from $2.5 \times 10^{-3}$ cm$^{-3}$ in the WCB outflow case, with a concurrent decline in $r_{ice}$ from 15 µm to 14 µm (Figures 3b and 6b). The effect on the WCB outflow case, which has lower ICNC and larger ice crystals to start with, is comparatively lower when considering the median values of ICNC and $r_{ice}$.

This shift in particles size in both clouds is also seen in Figure 7, which shows that in both cases, the ICNC×2 experiments have higher ICNC, IWC, RH$_{ice}$ and temperature after perturbation, while mean $r_e$ decreases. As shown in Figures 1 and 5, ICNC is initially higher in the GW case, so doubling has a larger effect in absolute terms than in the WCB outflow case on all parameters shown. Figure 7 and Table 2 show that median perturbed ICNC in the stable phase of the GW ICNC×2 case is 50% higher than in the control, with maximum differences reaching approximately 1 cm$^{-3}$. This compares with a median and maximum difference of $2.1 \times 10^{-3}$ cm$^{-3}$ and approximately 0.1 cm$^{-3}$, respectively, between the control and ICNC×2 simulations in the WCB outflow case, i.e. ICNC is approximately two times higher in the perturbed simulation.

While ICNC is higher in the perturbed simulations by design, on first consideration the positive IWC differences (median and maximum differences of $7.0 \times 10^{-4}$ mg m$^{-3}$ and approximately 0.6 mg m$^{-3}$ respectively, in the stable phase of the WCB outflow case, and 0.34 mg m$^{-3}$ and approximately 4 mg m$^{-3}$ respectively, in the stable phase of the GW case) shown in the top panels of Figure 7 and in Table 2 may seem counter-intuitive, as no additional source of ice is supplied in the perturbations (recall that IWC stays the same during the perturbed ICNC experiments). However, as shown in Figure 5 and Table 2, the primary effect of perturbing ICNC in these idealised simulations is to extend cloud lifetime. Consequently, for a given time in each simulation, the control simulation is more developed and has a larger IWC than experiments with a positive ICNC perturbation applied.







**Figure 7.** As in Figure 1, except the colour contours now show the difference between the ICNC×2 and the control simulations for the gravity wave cirrus (left) and the warm conveyor belt outflow cirrus (right). Red colours indicate that the perturbed simulation has larger values than the control, while blues indicate smaller values.



This size-driven effect modifies in-cloud thermodynamical properties too. Because the cloud retains more ice in the ICNC×2 simulations than the control, in-cloud $RH_{ice}$ stays higher (median and maximum positive differences, respectively, of 0.8% and approximately 1.4% for WCB outflow and 0.06% and approximately 7% for GW). Temperature differences are very slightly positive (mean differences of <0.01 K and 0.02 K for the WCB outflow and GW cases, respectively) because the vapour deposition process causes a small amount of latent heating of the atmosphere during the conversion from water vapour into ice.

### 3.6 Impact of idealised perturbations on cloud microphysical processes

To explore further the microphysical responses to the perturbations, we now consider in-cloud microphysical process rates. Figure 8 shows the mean differences between the largest process rates in the control and ICNC×2 perturbation simulations for the GW and WCB outflow cases. In both cases, vapour deposition is the primary mode by which newly nucleated ice particles grow, and deposition rates in the perturbed cases are larger than in the control simulation because the more numerous, smaller ice particles have a greater surface area onto which water vapour can be deposited. For example, deposition rates in the GW and WCB outflow cases reach maximum values of 9.3 and 0.37 mg kg$^{-1}$ s$^{-1}$ by the end of the control simulation, respectively. Similarly, there is a greater surface area from which to lose water vapour via sublimation, hence the perturbed simulations also feature higher sublimation rates than the control simulations in the layer that contains most ice particles (the red areas in the sublimation panels in Figure 8). In both cases, the area of positive sublimation differences, with mean values of 1.8 and 0.03 mg kg$^{-1}$ s$^{-1}$ in the GW and WCB outflow cases, respectively, is underlain by a layer with lower sublimation rates compared to the control (on average -2.3 and -0.04 mg kg$^{-1}$ s$^{-1}$ for GW and WCB outflow, respectively) because ice crystals are smaller, so there is a time delay in their descent towards the cloud base, which shows as dipoles in the distribution of changes in sedimentation rates. As shown in Figure 1 for the control cases, saturation is lower at the cloud top ($RH_{ice}$ of ≈ 100%) and ICNC and IWC are consequently lower there than in the centre of the cloud. Hence, ICNC perturbations have a smaller effect on overall cloud ice mass at the cloud top (Figure 7). However, Figure 7 shows that in the WCB outflow case particularly, $r_e$ is reduced throughout the entire vertical extent of the cloud compared with the control. So, despite having ICNC and IWC that are approximately the same as in the control, the top layer of the perturbed WCB outflow cloud has smaller ice crystals that sediment at a slower rate compared to the control (Figure 8).

Differences in growth rate, which is calculated as the sum of ice mass fluxes from the aggregation and autoconversion processes (the transition from ice crystals to snow), are largest at the base of the cloud where the largest ice particles are concentrated. Differences are initially negative compared to the control when fewer of the ice particles in the perturbed cloud reach snow sizes, and then become positive as ice and snow crystals in the control simulation reach critical size to precipitate, while those in the perturbed simulation stay in the cloud. This represents a displacement of the maximum growth rates later in time in the perturbed simulation, consistent with its prolonged lifetime.





510

**Figure 8.** As in Figure 7, except for microphysical processes: water vapour deposition, growth (which represents the sum of growth via aggregation + autoconversion), sublimation and sedimentation. Note that all terms include process rates for ice + graupel + snow.





### 3.7 Sensitivity of simulated cirrus to idealised perturbations

To quantify the impact of idealised perturbations on the cirrus water budget, we examine the sensitivity of cloud IWP to ICNC perturbations. Figure 9 shows the sensitivity of both cirrus types to the ICNC perturbation imposed, where the sensitivity is defined as $\Delta ln(IWP)/\Delta ln(ICNC)$. $\Delta ln(ICNC)$ is taken at the moment when the perturbation is applied and is therefore equal to the natural logarithm of the imposed perturbations ($\times 0.5$, $\times 0.9$, etc.). $\Delta ln(IWP)$ is calculated as the difference in IWP between the perturbed and control simulations, taken 30 and 45 minutes after applying the ICNC perturbation, i.e. 2,000 s and 3,000 s. These times are within the response phase of the cirrus, equivalent to the stable phase that we defined above for the control simulations, but for the perturbed simulations (see Figure S2 and discussion in the supplementary materials).

Figure 9 shows that there is a linear IWP response for smaller perturbations (between ICNC$\times 0.9$ and ICNC$\times 1.5$). The larger perturbations, namely the ICNC$\times 10$, ICNC$\times 2$, ICNC$\times 0.5$ and ICNC$\times 0.1$ experiments, depart from this linear behaviour (as shown in Figure S3 for the ICNC$\times 10$ case), with an IWP response smaller than predicted by the linear fit. This is as expected from non-linearities in water vapour availability and sedimentation velocities. Sensitivities are therefore calculated by fitting the IWP response over that linear range, excluding the more extreme ICNC$\times 0.1$ and ICNC$\times 10$ experiments where non-linearities appear. Figure 9 shows that the WCB outflow case is more sensitive to a change in ICNC compared to the GW case, probably because its initial ICNC and IWP are much smaller than in the GW case. This translates as larger slope values for the WCB outflow case (m=0.22 and m=0.35 after 30 and 45 minutes, respectively), as shown in Table 3. The intercept of the line of best fit for both cases is close to zero, as expected. Figure 9 also shows that the sensitivity increases with time, with increasingly large differences in IWP between the perturbed and control simulations, consistent with the results presented in Figure 5. This is because the control cirrus dissipates and loses ice mass sooner with increasingly positive ICNC perturbations and loses mass later with negative ICNC perturbations.

**Table 3.** Sensitivity, $\Delta ln(IWP)/\Delta ln(ICNC)$, 30 and 45 minutes after perturbing ICNC for the gravity wave cirrus and warm conveyor belt outflow cirrus cases. m and c denote the slope and intercept of the lines fitted to the IWP response to ICNC perturbations, respectively.

| Time | Gravity wave | | Warm conveyor belt | |
|---|---|---|---|---|
| | $m$ | $c$ | $m$ | $c$ |
| 30 minutes | 0.03 | −0.0005 | 0.22 | −0.003 |
| 45 minutes | 0.06 | −0.0012 | 0.35 | −0.010 |





**Figure 9.** Sensitivity of ice water path (IWP, in g m$^{-2}$) to prescribed perturbations to ice crystal number concentrations (ICNC, in m$^{-3}$), expressed as differences in their natural logarithm between perturbed and control simulations. The gravity wave (GW) cirrus is in red, the warm conveyor belt outflow (WCB) cirrus in blue. Circles represent values 30 minutes after perturbation, squares represent values after 45 minutes. The coloured dashed-dotted and dashed-double dotted lines show lines of best fit for the most linear part of the response.



## 4 Conclusions

In this study, we use the MONC LES model to simulate a gravity wave cirrus cloud and a warm conveyor belt outflow cirrus cloud. The cloud macro- and microphysical properties are within the expected ranges for these two types of cirrus clouds and
close to the averages in their respective temperature ranges, although not representative of average cirrus clouds. The two simulated cirrus are then perturbed by abruptly increasing their ICNC to mimic, in an idealised way, the effect of an aircraft-emitted aerosol plume reaching an existing cirrus cloud.

The primary impact of these idealised ICNC perturbations is to change the mean size of the simulated particles, therefore increasing the lifetime of the cloud. That is because increasing ICNC while keeping IWC constant distributes cloud water content across a larger number of ice particles, meaning cloud ice particles become smaller. These smaller ice particles take longer to grow via vapour deposition and accretion, and longer to sediment out of the cloud, thus extending cloud lifetime. The magnitude of the effect is proportional to the magnitude of the perturbation; that is, larger ICNC perturbations extend the lifetime of the cloud further relative to the control simulations. The responses also develop in time, with IWP changes being larger after 45 minutes than after 30 minutes. The effect is qualitatively the same for both the GW and WCB outflow cases, although the magnitude of the effect was smaller for the WCB outflow case, which had lower ICNC and IWC to begin with. The IWP sensitivity of both clouds can be quantified by the ratio $\Delta ln(IWP)/\Delta ln(ICNC)$ of their perturbation, which are 0.06 and 0.35 for GW and WCB outflow, respectively, 45 minutes after the ICNC perturbation.

This work explores the end of the aerosol-ICNC-IWP chain along which a perturbation in INP number would propagate. We do not explicitly examine aerosol activation or contrail formation in this study. Therefore, more work is required to take this analysis one step further and explore the first half of the chain, from aerosols to ice crystals, and whether representing that first half has an influence on the IWP sensitivities described here. Kärcher et al. (2021) suggest using 1D LES modelling that the first half of the chain from aerosol to ICNC perturbation is not straightforward. They find that adding soot aerosol into their simulations actually forms clouds with fewer, larger particles because if soot aerosols act as efficient INP they suppress homogeneous freezing. The IWP response would then follow the pathway that corresponds to our simulation where ICNC is decreased.

This study examined only two types of cirrus, the GW cirrus that occurs relatively infrequently in the atmosphere according to Krämer et al. (2020), and the WCB outflow cirrus, which is more representative of background cirrus. Further exploration is required to extend our conclusions to other types of cirrus cloud and other meteorological conditions, especially in terms of updraft velocities, which play a crucial role in determining ice crystal formation rates and sizes.

**Acknowledgements**

This work was funded under the European Commission Horizon 2020 Advancing the Science for Aviation and ClimAte (ACACIA) project (grant agreement ID 875036). EG and NB acknowledge the use of the MONSooN system, a collaborative facility supplied under the Joint Weather and Climate Research Programme, a strategic partnership between the Met Office and the Natural Environment Research Council. In addition, NB and JP acknowledge support from the French Ministère de la Transition Ecologique et Solidaire (grant no. DGAC N2021-39), with support from France's Plan National de Relance et de Resilience (PNRR) and the European Union's NextGenerationEU. We are also grateful to Adrian Hill for invaluable support in setting up the MONC model, to Peter Spichtinger for generously supplying radiosonde data for the warm conveyor case, and to Julien Karadayi for his assistance in producing the model simulations.

**Data availability**

Initial data for the gravity wave case can be found at http://homepages.see.leeds.ac.uk/~lecsjed/huiyi/gcss/ (accessed 3 April 2021) and ERA5 data used to derive initial profiles for the warm conveyor belt case are available via the Copernicus Climate Data Store (https://cds.climate.copernicus.eu/#!/home). Model simulation data are available as Gilbert, Purseed & Bellouin (2024) at https://doi.org/10.5281/zenodo.10845637.

**Author contributions**

EG and NB designed the research. EG and JP performed the simulations. YL and MK provided insights and links to aircraft measurements. EG led the writing of the paper, to which all authors contributed with text and comments.

**Code availability**

Code for the Met Office NERC Cloud (MONC) large eddy simulation model used in this study is available after registration at the UK Met Office.

**Competing interests**

MK is on the editorial board of *Atmospheric Chemistry and Physics*. The other authors declare that they have no competing interests.



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
