# Peer review of "Response of cirrus clouds to idealised perturbations from aviation"

_EGUsphere, 2024_

## Referee Comment (RC1)

Review of research article manuscript egusphere-2024-821

"Response of cirrus clouds to idealised perturbations from aviation"

submitted by Ella Gilbert et al.

This study quantifies the impact of ice crystal number concentration (ICNC) perturbations due to aviation activity on the water budget and microphysics of cirrus clouds. For this purpose, a large eddy simulation (LES) model was set up and initialized to follow the evolution of (perturbed) cirrus over six hours. Two cirrus types were studied at temperatures in the range 224-230 K that are subject to updraft speeds in the range 1-7 cm/s. Variations in imposed updraft speed caused a range of ICNC perturbations for each type of cloud. A sensitivity parameter was inferred as the ratio of simulated relative ice water path changes over ICNC changes.

Effects of cloudiness changes due to aviation have mostly been studied based on low resolution (global) climate models, in which associated microphysical processes (e.g., ice nucleation) and underlying dynamical forcing (e.g., gravity wave activity) are only crudely parameterized. The present study uses LES in an idealized framework aiming at better understanding aviation aerosol-related changes in cirrus properties. The manuscript is well-written and the analysis of the simulations is exemplary, however, I found aspects of the methodology and approach to be unclear or questionable, leaving a central claim of the study unsupported. These issues should be resolved before the manuscript can be accepted for publication in ACP.

**General comments**

The authors address in their introduction two possible mechanisms that may lead to cloudiness changes due to aviation that I briefly recall and comment on in more detail below:

> A. Aircraft emitted black carbon soot (BC) may impact the ice formation process in cirrus clouds. Numerous studies have shown that aviation BC particles are unlikely to nucleate ice heterogeneously in the upper troposphere (UT) and that their ice activity can be enhanced by processing in contrails. Even after such processing, they are poor ice nucleating particles (INPs).

> B. A significant amount of flights in the UT lead to the formation of contrails. Besides forming in outside of clouds, contrails can also form within pre-existing cirrus clouds or natural cirrus forms around contrails. Contrail ice formation in aircraft exhaust plumes involves aircraft BC particle emissions and differs from aerosol-induced cirrus ice formation in the UT.

Both mechanisms are unrelated and perturb cirrus clouds in their own characteristic ways depending on whether a new cloud is formed or one already exists.

1. Attributing aviation activity to simulated cirrus perturbations.

The authors describe the main purpose of their study (lines 20-22): "Here, we use a large eddy simulation model to quantify the impact of ice crystal number concentration (ICNC) perturbations on the water budget and microphysics of pre-existing cirrus clouds. These perturbations aim to represent the second half of the chain of effects linking aircraft aerosol emissions to changes in ICNC and ice water path."

Thus, the authors address mechanism A, but on the basis of a pre-existing cirrus cloud. This raises the question whether it is reasonable to assume that (line 555-556) "… cirrus are then perturbed by abruptly increasing their ICNC to mimic, in an idealised way, the effect of an aircraft emitted aerosol plume reaching an existing cirrus cloud."

While I understand that aviation perturbations are idealized in the model (which I do not cast into doubt in principle), I have strong reservations about the model perturbations representing real-world aviation activity. Perturbing "instantaneously across the whole model domain and throughout the entire profile" (line 184) is similar to what is done in coarser models in which clouds are not resolved. Instead, aerosol-cloud interactions occur locally and in particular, an aviation soot layer would perturb only a small atmospheric region (e.g., in the vertical) so that the association of the model perturbation with aviation activity (or any kind of aerosol-cloud interaction for that matter) appears to be unrealistic.

Cirrus perturbations as represented in the present LES model do not mimic mechanism A, because when an aircraft exhaust plume containing contrail-processed BC particles reaches already existing cirrus, it will not lead to a significant change in cloud properties. This is because pre-existing cloud ice will prevent, via rapid quenching of ice supersaturation, new ice nucleation. This holds true especially for aircraft BC particles that only nucleate ice at high supersaturation. New ice formation may occur only in regions where ICNCs stay below a few per liter of air for the updraft speeds considered in the manuscript.

Moreover, mechanism A cannot simply be replicated by "applying a multiplication factor to their distributions at the end of the spin-up phase" (line 183). A change in cirrus ICNC due to aviation soot-cirrus interaction involves additional changes in ice crystal size distributions and in the magnitudes of other cloud variables (e.g., ice water path) that do not simply scale with ICNC according to detailed simulations (Karcher et al., 2021). The latter study also suggests that the effect depends on where the aircraft BC particles are located in the vertical relative to ice supersaturation maximum and at which after emission the perturbation occurs.

The cirrus perturbations do not mimic mechanism B as well, because when an aircraft passes through a pre-existing cloud, it may generate a pencil-shaped (contrail) volume containing ice crystals which much higher number concentrations and much smaller sizes than present in the cirrus. The resulting bimodal ice crystal size distributions change microphysical processes locally. Such changes cannot be captured by scaling pre-existing cirrus ICNC instantaneously with a constant factor.

While I agree in principle that for the sake of conceptualization (line 114): "The aerosol-cloud interaction problem can be divided into two steps.", deliberately excluding the first half of the aerosol-ICNC-IWP chain in the present study prevents a proper treatment of the cirrus perturbation due to aircraft BC particles. Arguably, in order to treat mechanism A realistically, the aerosol impact has to be modeled explicitly. As implied on line 573, the study of Karcher et al. (2021) suggests that shortly after the aircraft BC perturbation, $\Delta\ln(IWP)>0$ and $\Delta\ln(ICNC)<0$, which invalidates the statement on line 576f: "The IWP response would then follow the pathway that corresponds to our simulation where ICNC is decreased." This means that realistically simulated aviation soot-related perturbations of cirrus cloud formation are not captured by linear $\Delta\ln(IWP)$-$\Delta\ln(ICNC)$ relationships, since the prescribed ICNC-perturbation of pre-existing cirrus can only yield $\Delta\ln(IWP)<0$ for $\Delta\ln(ICNC)<0$ (Fig.9 and Table 3).

In view of the statement (line 571-573): "Therefore, more work is required to take this analysis one step further and explore the first half of the chain, from aerosols to ice crystals, and whether representing that first half has an influence on the IWP sensitivities described here.", it might be a good idea to acknowledge that the "first half of the aerosol-ICNC-IWP chain" has been studied previously and that

such work suggests it does matter for the second half. Otherwise, the notion is created that this is completely uncharted territory.

In summary, this challenges the fundamental assumption made in this study that the second half of the chain of effects linking aircraft aerosol emissions to cirrus changes can be treated independently of the first half of the chain and thus raises serious doubts as to the validity of the final statement in the abstract: "These results suggest that aviation has the potential to increase the lifetime and radiative effects of pre-existing cirrus clouds."

2. Model set-up, initialization and cirrus classification.

Both selected cases have some observational meteorological background, but appear to be too low in altitude and too warm to represent average conditions at (midlatitude) cruise altitudes. One case (Spichtinger et al., 2005) relates to ice supersaturation between 320-408 hPa, where aircraft don't even cruise. Yang et al. (2012) is not in the reference list so I cannot comment further on it, but what is written about this case study on lines 146-147 seems counterintuitive: a lee wave cirrus should have encountered rather strong lifting and should initially contain many ice crystals with ICNCs at the upper end of observed values.

According to long-term in-situ measurements in the extratropical northern hemisphere, temperatures at cruise altitudes are in the average range 214-224 K; values >226 K are rare even at the lowest flight levels (down to 270 hPa). It is unclear how the choice of meteorological case studies affects the representativeness of the results presented in the study. At any rate, the synoptic analyses underlying both cases may not capture the full gravity wave (GW) impact, especially the high frequency components in the vertical wind field that are crucially important for ice nucleation in cirrus are likely missed.

Regardless of the meteorological set-up, the distinction between cirrus types (line 99): "Cirrus can further be classified according to vertical velocity as either slow (<10 cm/s) or fast (>10 cm/s) updraft cirrus" makes little sense to me. This is because the lifetimes of the simulated cirrus (6 h, Fig. 8) or even the times past cirrus formation after which sensitivity parameters are evaluated (30-45 min, Fig. 9) well exceed the time intervals over which updraft speeds generated by high frequency GWs are coherent, typically a fraction of a buoyancy period (Podglajen et al., 2016). During and after formation, cirrus clouds are subject to a very wide range of rapidly fluctuating vertical wind speeds superimposed on much slower variability.

Thus, the W=10 cm/s updraft speed separation appears to be an arbitrary choice, and it remains unclear why it is introduced in the first place. Updraft speed standard deviations due to GW lie in the range 10-20 cm/s. The values of W listed in Table 2 are therefore not representative for GW activity. These may rather represent a slowly variable (background) updraft onto which GW variability is to be superimposed. According to in-situ measurements, mesoscale vertical wind variability is the primary cause of broad ICNC distributions within cirrus clouds.

The method of creating pre-existing cirrus (lines 223-224) during the relaxation of the model spin up phase is obviously a numerical experiment that does not guarantee UT ice formation processes to be reliably replicated, due mainly to the lack of realistic dynamical forcing as noted above. Since ICNC distributions of the resulting cirrus are key to the present study, it would be helpful if the authors could produce from their Fig.3 probability distributions of ICNC (perhaps during the stable phase) that may be compared with field data of midlatitude cirrus representative for cruise conditions (e.g., Karcher and Strom, 2003; Hoyle et al., 2005; Jensen et al., 2013).

In that regard, (lines 141-142): "We emphasise that this method of initialising the model cloud is not necessarily reflective of the actual ice nucleation mechanisms of the two cirrus cases studied." introduces an inconsistency that may be consequential for the simulated cloud evolution that remains to be explored.

All in all, this challenges the representativeness of the selected cirrus cloud cases for deriving the sensitivity parameter shown in Fig.9, again questioning the validity of the statement (lines 28-29): "These results suggest that aviation has the potential to increase the lifetime and radiative effects of pre-existing cirrus clouds."

**Other comments**

i. "Parameterisations of updraft velocity are especially important in the kind of large-scale models that are typically used to make global estimates of the radiative effect …" (lines 68ff). I agree, but such parameterizations would also be needed in the LES framework. Please refer to general comment 2.

ii. "… whereas those assuming soot is an inefficient INP show much smaller effects (Gettelman & Chen, 2013; Kärcher et al., 2021)". The assumption holds for the climate model. By contrast, the latter study did not just "assume" soot to be an inefficient INP, but based the microphysical simulations on a physically-based parameterization of the nucleation mechanism constrained by laboratory data. They did find a non-negligible reduction of nucleated ice crystal number concentrations due to the ability of aviation soot to weaken homogeneous freezing events. Please clarify.

iii. The introduction mentions mechanism B on lines 104-112. This confused me and only after a second reading I understood that this study does not deal with contrail formation (within cirrus). Is it necessary to elaborate on mechanism B? If so, it should be clearly distinguished from A to avoid confusion.

iv. "The perturbations induced are intended to be plausible." (line 119). I don't think so. Please refer to general comment 1 above.

v. Please state explicitly why / how the domain experiences an updraft (see W-entries in Table 2).

vi. How realistic is a water vapour forcing applied over 3 km depth in a subsidence region (lines 156-159)? Why would an otherwise dissipating cirrus be representative for aviation perturbation estimates?

vii. The authors chose a larger vertical than horizontal resolution (line 131). I wonder whether this choice is compatible with e SGS turbulence scheme in the LES model and which effect the rather coarse vertical resolution of 120 m has on the treatment of sedimentation?

viii. On which grounds do you decide that the "ICNC×0.5 and ×2 experiments represent large but still-plausible perturbations" (line 190)? Please refer to my comment on mechanism B above (general comment 1).

ix. "Homogenous nucleation rates are relatively higher at the cloud base, whereas heterogenous nucleation is comparatively higher at cloud top." on lines 229-230 sounds counterintuitive. Does the nucleation parameterization scheme include competition between different nucleation modes for available water vapor during ice formation?

x. The statement on lines 575-576: "Kärcher et al. (2021) suggest using 1D LES modelling that the first half of the chain from aerosol to ICNC perturbation is not straightforward. They find that adding soot aerosol into their simulations actually forms clouds with fewer, larger particles because if soot aerosols act as efficient INP they suppress homogeneous freezing." does not reflect results of that study, namely that aircraft BC particles are poor INPs only forming ice alongside homogeneous freezing of supercooled solution droplets, even under conditions that maximize their impact. Please correct.

**Overall recommendation for revision**

Besides considering and clarifying the above issues, I recommend the authors remove the inference (line 29): "These results suggest that aviation has the potential to increase the lifetime and radiative effects of pre-existing cirrus clouds." I strongly suggest to remove (not weaken), since the way cirrus perturbations are treated have no link to aviation activity whatsoever that could be justified on a physical basis, plus the perturbed cirrus properties and dynamical regime may not be capturing cirrus types and small-scale meteorological conditions prevalent at cruise altitudes.

Importantly, the above criticism extends to the Short Summary, stating: "We show that the main effect of our experiments – which intend to mimic the effect of aircraft soot emissions reaching existing high-altitude cirrus clouds – is to extend cloud lifetime, thereby enhancing their effect on climate." This far-reaching statement cannot be maintained and I strongly recommend to remove it as well.

The perturbation of a pre-existing cirrus cloud, in which cloud properties are abruptly changed over the whole cloud domain does not harvest the full potential of what the high-resolution LES framework employed in this study may actually accomplish. It is necessary to explicitly discuss issues with relating the idealized perturbation to aviation activity. Even an idealized treatment should be reasonably realistic. What would be its real-world equivalent?

Concerning the discussion of sources of uncertainties in the introduction, it may be worth pointing out that there is convergence in experimental studies (both lab and ground-based measurements) showing that aircraft BC particles are poor INPs when their small size (< 50-100 nm) is considered. This stems mainly from morphological considerations (BC particle nanopore geometry after cloud processing) which should apply and thus allow extrapolation to UT conditions. None of the quoted climate model studies represents the competition between aircraft BC particles and liquid solution droplets based on the latest information from the lab, let alone includes the competition with other, more potent INPs such as mineral dust, see Karcher et al. (2023).

Besides covering the many meteorological conditions in which various cirrus types form and evolve (lines 580-582), a major future challenge is the explicit treatment or parameterization of aerosol-induced cirrus ice formation in a limited-area LES framework along with realistic gravity wave forcing. Maybe this is worth pointing out in the conclusions.

**References**

Karcher, B. and Strom, J.
The roles of dynamical variability and aerosols in cirrus cloud formation.
ACP 3, https://doi.org/10.5194/acp-3-823-2003 (2003).

Hoyle, C. R., Luo, B. P. and Peter, T.
The origin of high ice crystal number densities in cirrus clouds.
JAS 62, https://doi.org/10.1175/JAS3487.1 (2005).

Jensen, E. J., Lawson, R. P., Bergman, J. W., Pfister, L., Bui, T. P. and Schmitt, C. G.
Physical processes controlling ice concentrations in synoptically forced, midlatitude cirrus.
JGR 118, https://doi.org/10.1002/jgrd.50421 (2013).

Podglajen, A., Hertzog, A., Plougonven, R. and Legras, B.
Lagrangian temperature and vertical velocity fluctuations due to gravity waves in the lower stratosphere.
GRL 43, https://doi.org/10.1002/2016GL068148 (2016).

Karcher, B., Marcolli, C., and Mahrt, F.
The role of mineral dust aerosol particles in aviation soot-cirrus interactions.
JGR 128, https://doi.org/10.1029/2022JD037881 (2023).

---

## Referee Comment (RC2)

Review of "Response of cirrus clouds to idealised perturbations from aviation", by Gilbert and coauthors, EGUSphere-2024-821

I have carefully gone through the manuscript and am very familiar with aircraft contrails and contrail development, along with the associated properties of contrails. I have given considerable thought to the question of how aircraft passage through a cirrus cloud is affected by the passage and have discussed the effects with researchers at several institutions, using data collected by aircraft and satellite-borne sensors. I appreciate the time and effort taken by the authors of this article, and the LES model used to study the potential effects of the addition of ice nuclei added by the aircraft during its passage. The finding that gravity waves have considerably more effect than warm conveyor belt outflow on the resulting ice water path is interesting.

I have numerous comments in my review of the manuscript by I'm left with a concern that I feel makes the manuscript incomplete in its current form. The exhaust from combustion by the aircraft engines, and the wake turbulence from the aircraft, affects the downstream heat and moisture, along with the dynamics of the air. I note this from the Wikipedia article passage below. The vortices sink at a rate of 3 m/s or more and stabilize at about 150-270 m below the aircraft. As a result, the pre-existing cirrus crystals might sublimate rather at the aircraft level rather than the additional generation of copious ice crystals generated from the combustion products. In fact, for the thinner cirrus, holes (Distrails) may be produced (see below). The article by Marjani et al. (2022) shows this effect clearly at the flight level.

To summarize, I feel that the use of the LES model, without incorporating the effects of the aircraft combustion and wake turbulence makes the study incomplete and in some cases may lead to the opposite effect. I hope the authors can take these effects into account in a revised version of the article.

Wikipedia Wake Turbulence

The vortex circulation is outward, upward, and around the wingtips when viewed from either ahead or behind the aircraft. Tests with large aircraft have shown that vortices remain spaced less than a wingspan apart, drifting with the wind, at altitudes greater than a wingspan from the ground. Tests have also shown that the vortices sink at a rate of several hundred feet per minute, slowing their descent and diminishing in strength with time and distance behind the generating aircraft.[2]

At altitude, vortices sink at a rate of 90–150 m (300–490 ft) per minute and stabilize about 150–270 m (490–890 ft) below the flight level of the generating aircraft. Therefore, aircraft operating at altitudes greater than 600 m (2,000 ft) are considered to be at less risk.[3]

[Figure]

Figure 1, Marjani, S., Tesche, M., Bräuer, P., Sourdeval, O., & Quaas, J. (2022). Satellite observations of the impact of individual aircraft on ice crystal number in thin cirrus clouds. *Geophysical Research Letters*, 49, e2021GL096173

Where an aircraft passes through a cloud, it can disperse the cloud in its path. This is known as a distrail (short for "dissipation trail"). The plane's warm engine exhaust and enhanced vertical mixing in the aircraft's wake can cause existing cloud droplets to evaporate.

[Figure]

**Distrails & Rain** ~ The path of an aircraft in high cirrus is marked by a dark trail, a distrail or dissipation trail. Its formation is the result of subtle differences between ice and supercooled water. Images taken in Germany by Sebastian Luft. Images ©Sebastian Luft , shown with permission

---

## Referee Comment (RC3)

**Review of "Response of cirrus clouds to idealised perturbations from aviation" by Ella Gilbert et al., submitted to Atmospheric Chemistry and Physics (ACP)**

**[MS No.: egusphere-2024-821]**

This study uses a large eddy simulation model to quantify the impact of ice crystal number concentration (ICNC) perturbations on the water budget and microphysics of pre-existing cirrus clouds. It examines two specific types of cirrus—gravity wave cirrus and warm conveyor belt outflow cirrus—and their responses to simulated perturbation in ICNC, which are intended to represent the effects of aerosol emissions from aircraft. The research finds that higher ICNC extend the cloud's lifetime by reducing the size of ice particles, which then take longer to grow and sediment out of the cloud. This effect is more pronounced in the gravity wave cirrus case. Additionally, the study assesses the sensitivity of the ice water path (IWP) to these perturbations, noting that the degree of impact varies based on the type of cirrus cloud and its initial conditions of ICNC.

I appreciate the authors' effort in utilizing the LES model for this study and their examination of how cloud microphysical processes evolve during perturbations over time. The paper is engaging, and its language makes it accessible and straightforward to follow. The descriptions of the simulations are both clear and concise. However, I feel that the paper does not fully deliver on what the title promises. The methodology and assumptions used to assess the impact of aviation on ICNC changes in cirrus clouds require substantial revision. Therefore, it seems inappropriate to link the study's conclusions to the effects of aviation, even if idealized. Consequently, I recommend that this paper undergo a major revision before it can be considered for publication in ACP.

**General Comments**

**1. Environmental Suitability for Contrail Formation**

Before studying the effects of contrails on cirrus clouds, we must first verify whether contrails will form under the given environmental conditions or not. The Schmidt–Appleman criterion provides the threshold temperature, representing the warmest possible conditions conducive to contrail formation. This criterion depends on ambient air pressure, humidity, and specific aircraft characteristics. Additionally, contrails will only persist if the ambient humidity is at least saturated relative to ice.

I highly recommend that the author check and demonstrate whether the cloud cases are appropriate for contrail formation before analyzing their main results. I question, especially, the suitability of the GW case described in lines 150-154 for studying the impact of contrails, as it is characterized as "a relatively

warm and low-altitude cirrus cloud, with a mean modeled in-cloud temperature of 229 K and formation occurring at 8.5 – 9.5 km altitude, with a supersaturation of ~10% with respect to ice." However, this 110% is at the start of the simulation, and for realistic calculations, the author should use the supersaturation at the time they applied the perturbation, which is at the end of the spin-up phase at t=1500 s in the GW case. Perhaps the information in Figure 5c is more relevant, which clearly shows that in-cloud relative humidity over ice is around 100%-100.3% (also mentioned in lines 375-377).

Given that the top of the atmosphere is at 22.5 km, it can be inferred that the case is assumed to be in a mid-latitude region. Comparing this with two relevant studies, Verma & Burkhardt (2022) indicate that typical cruise levels in Germany range between 10.3 km and 10.8 km, where the ambient temperature often falls well below the contrail formation threshold. Additionally, Li et al. (2023) indicates that the most frequent aircraft cruising altitudes correspond to a pressure range of 200–245 hPa (207–218 K in temperature). Based on the provided discussion, I strongly believe that the GW case does not meet the Schmidt-Appleman criterion for contrail formation and is not a suitable case for the purpose of this study.

**2. Contrail-Cirrus Cloud Interactions and Methodological Approaches**

In studying contrail-cloud interaction, it's crucial to consider several key factors. First, the ice crystals from contrails are notably smaller than those in natural cirrus clouds, due to the combustion of fuel which releases limited water vapor and high number of nucleated ice crystals. Consequently, contrails tend to introduce smaller ice crystals into cirrus clouds, creating perturbations that are not accurately represented by merely multiplying the ice crystal number concentration of natural cirrus clouds, which typically feature a different mass mean diameter (Voigt et al., 2017; Unterstrasser, 2017; Verma & Burkhardt, 2022).

Furthermore, the number of ice crystals formed during contrail formation varies based on aircraft and fuel characteristics, influenced by factors like the release of aerosol particles and the atmospheric conditions (Kärcher et al., 2015). When an aircraft passes through a cloud, it introduces (if the persistent contrail forms at all) additional ice crystals into the atmosphere based on these parameters, suggesting an increase in ICNC by "addition" rather than by "multiplication" of the initial concentration which is used in this study.

Additionally, the method introduced in the paper, which involves multiplying the initial concentration of the cirrus cloud by various scale factors, sets another issue. By employing this approach, we essentially assume that when the ice crystal concentration of a non-perturbed cloud is higher, then the aviation perturbation is also stronger. This assumption may affect the interpretation and comparisons of the cases in the results.

Another important consideration is the interaction between the high concentration of small ice crystals in contrails and the relatively lower concentration of larger ice crystals in natural cirrus clouds. The dynamic interplay between these two groups of ice crystals is crucial and yet is overlooked in this study.

**3. The Need for Reassessing ICNC Perturbations**

After reviewing the findings from Marjani et al. (2022), I've noticed several critical arguments in their paper that seem to be overlooked in the current manuscript regarding the selection of ICNC perturbation values based on their result. The ICNC data in Marjani et al. was derived from DARDAR-Nice retrievals, which accounts only for ice particles larger than 5 micrometers. Considering that contrail-generated ice crystals are typically much smaller than those in natural cirrus clouds, the dataset from Marjani et al. (2022) predominantly represents those ice crystals larger than 5 micrometers. Consequently, adopting ICNC perturbation values based on these results without discussing or accounting for the 5 micrometer size threshold in your simulation is not a valid scientific assumption to proceed with.

**From Marjani et al. (2022) paper:** the mean diameter of ice crystals in young contrails (up to 1 hour) is typically smaller than 10 μm (Bock & Burkhardt, 2016b; Schröder et al., 2000). It is even less than in young cirrus clouds, which were found to be 10–20 μm (Schröder et al., 2000). **Therefore, it is possible that we have lost information about a certain fraction of contrail's ice crystals, those which are smaller than the retrieved threshold of 5 μm in the DARDAR-Nice product.**

**4. Demand for Statistical Accuracy in Median Change Analysis**

A Considerable part of the analysis in the manuscript is based on observing the median changes in various atmospheric variables like temperature, updraft speed, relative humidity, IWC, and ICNC between the control and perturbation simulations, labeling these differences as "large" or "small." However, the analysis doesn't really dig into the statistics to support these claims. To make the findings more solid, it would be essential to include tests for statistical significance and to share the uncertainty ranges for these median values. This approach will ensure that the observed changes are both statistically significant and relevant to the study's objectives.

**5. Magnitude of the effect vs. Sensitivity**

The paper mostly talks about the magnitude of the effect, especially highlighting how much more the GW case perturbations affect the Ice Water Path (IWP) compared to the WCB case. Understanding both how sensitive and how big these effects are is important to really get what's going on when things change in a system, but the paper doesn't really focus much on the sensitivity part (last section). It kind of sticks to talking about how big the effects are. Toward the end, though, it suddenly says that the WCB outflow might actually react more to changes in ICNC, but it doesn't do a good job of explaining or connecting this idea back to the earlier parts. This flip-flop between focusing on how big the effects are and then jumping to sensitivity without much explanation might leave readers a bit confused.

Additionally, I think it would make things clearer and more comparable if, especially in the sensitivity analysis, we compare the cases based on equal changes (the addition of constant amount of perturbation in

both cases instead of multiplication as discussed earlier in comment 2). This way, we can observe how each case responds to the same level of perturbation, making it easier to understand and compare their responses.

**Specific Comment**

**1. Abstract**
To enhance clarity and coherence, it would be beneficial for the authors to include a short explanation in the abstract of how this methodology is representative of aviation impact.

**2. Lines 25-26**
The sentence suggests that the more pronounced effect in gravity wave cirrus compared to warm conveyor belt outflow cirrus is due to the latter having lower initial ICNC and ice water content. However, this reasoning may need clarification. It's not necessarily the lower ICNC and IWC that directly cause the difference in effect between the two cases. **Instead, it could be attributed to the strength of the perturbation, which is stronger in the gravity wave case due to initial higher ICNC.** Therefore, it might be more accurate to specify that **the stronger perturbation in the gravity wave case, resulting from higher initial ICNC, leads to the more pronounced effect**. Clarifying this point would enhance the interpretation of the sentence.

**3. Lines 114-121**
In the final paragraph of the introduction, there's a lack of explicit explanation regarding the study's connection to aviation, similar to the abstract. Instead, the focus is primarily on the aerosol-cloud interaction, outlining the two steps involved in this process. While the authors mention examining the response of cloud microphysical processes, properties, and water budget to ICNC perturbations, which aim to mimic the addition of ice-nucleating particles (INPs) that freeze heterogeneously, the direct link to aviation is not clearly articulated. If the study aims to investigate the impact of aviation, it should be explicitly addressed in this section.

**4. Section 2.2 Case Description**
It would be beneficial to include a brief explanation in the case description section regarding the differences in model top height between the two cases (22.5 km and 15.9 km), the rationale behind applying water vapor forcing in one case but not the other, and the decision to apply wind forcing in one direction in one case and in both directions in the other. Providing this explanation will help readers better understand the model setup and the reasoning behind the choices made.

**5. Lines 182-186**
The explanation effectively outlines how the perturbation was incorporated into the model. However, it would be crucial to further elaborate on how these perturbations are representative of contrail perturbations within the cloud environment. Contrails contribute to the presence of high concentrations of small ice crystals. Therefore, it's inappropriate to represent contrail signatures simply by multiplying the

ice crystal number concentration (ICNC) in natural cirrus clouds, which usually have different mean mass size. (explained more in general comment 2)

**6. Lines 191-192**

Why were the IWC×2 and ICE×2 experiments only done for the GW case? The manuscript mentions around lines 299-301 that "Considering that WCB cirrus are common in mid-latitudes, a region that has a high density of air traffic, the impacts of increasing ICNC on WCB outflow cirrus may be considered more directly relevant for examining potential aviation-aerosol-cirrus interactions". Then again in lines 579-580 the manuscript mentions that "the GW cirrus occurs infrequently in the atmosphere". Then wouldn't it make more sense to dive deeper into the WCB case?

**7. Lines 229-230**

It's somewhat unusual to find higher rates of homogeneous nucleation at the cloud base and higher rates of heterogeneous nucleation at cloud top. Providing an explanation to this unusual vertical distribution of nucleation processes would enhance clarity.

**8. Lines 394-403**

The text mentions a median updraft speed increase from 3.34 to 4.63 cm/s, which is about a 38.6% rise, yet labels it as a minimal effect. Meanwhile, it describes changes in ice crystal mass mean radius — from 12.39 to 11.43 micrometers in the GW case and from 14.72 to 14.2 micrometers in the WCB case — as large differences. This inconsistent interpretation of changes could be confusing. The labeling of changes as large and small needs further clarification to enhance the analysis's consistency and credibility.

**9.** I suggest discussing the differences in $r_{ice}$ values with at least one decimal point to ensure precision. While rounding may not significantly impact the $r_{ice}$ differences in the GW case (e.g., from 12.39 to 11.43 vs. 12 to 11, which results in nearly 1 micrometer difference), it does affect the WCB case. Here, rounding 14.72 and 14.2 to 15 and 14 alters the difference from 0.52 to 1 micrometer. **This is especially relevant since the author describes these as large differences in line 400**.

**10. section 3.4**

I found the discussion on IWC perturbations in the GW case within Section 3.4 particularly interesting. However, it remains unclear how this section contributes to the main claims made in the paper's title regarding perturbations from aviation. Could the authors clarify the connection and explicitly detail how these IWC perturbations are representative of those caused by aviation activities?

**11. Figure 6 and Figure 3**

It appears that Figure 6 is identical to Figure 3. I request the author to check if the same figure was mistakenly uploaded twice.

**12. Lines 455-457**

The concern mentioned here is crucial, yet its placement feels somewhat out of context. I was surprised it wasn't emphasized earlier, especially during the discussion of Figure 5 in Section 3.3. It is important to frequently remind readers that **the quantity of doubling in the GW case represents a significantly**

**stronger perturbation compared to the doubling in the WCB case**. This should also be taken into account when analyzing the results.

**13. Lines 477-479**

The text mentions that in-cloud $RH_{ice}$ remains higher in ICNC x2 simulations compared to control, yet it is unclear where these values are obtained from. I could not find them in Table 2 or Figure 5. If the mentioned medians are from Table 2, then GW case shows a **decrease** of 0.06%, and WCB case shows an **increase** of 0.8%. Could the author please clarify the source of the values in these lines?

**Minor Comment**

**1. Lines 21-22**

In the abstract, it says the study focuses on **'the second half of the chain'** but doesn't really explain what this 'chain' is all about. Later, around lines 114-116, the paper does a good job explaining this chain as a two-step process related to how aerosols interact with clouds. It might help if the abstract gives a clearer explanation of this chain to make things easier to grasp for the reader.

**2. Line 45**

It would be helpful to cite specific examples or studies that illustrate these discrepancies.

**3. Lines 139-140**

The author could enhance the flow of the sentence by considering a slight refinement. For instance, 'In the model, at simulated cirrus temperatures,' could be replaced with 'At the cirrus temperatures simulated'.

**4. Reference**

The reference for Yang et al. (2012) seems to be missing from the reference list. It is recommended that all references cited in the text be included in the reference section.

**5. Line 159**

The given unit includes an extra "$kg^{-1}$"

**6. Figure 1**

It is suggested that the time in the figure be presented in seconds rather than hours, as mentioned in the accompanying text. This adjustment would facilitate comprehension for the reader and consistency within the paper.

**7.** Please ensure consistency in the notation used for variables like "$r_{ice}$" and "$R_{ice}$" . Additionally, maintain uniform terminology across the manuscript, tables, and figures for clarity.

**8.** In figures 3 and 6, it is recommended to include the cloud type in the figure title, as done in figures 1, 2, 5, and 7.

**9.** In Figure 5, please ensure consistency in the notation used for 'mmr-weighted' and 'MMR-weighted' between the y-axis title and the caption.

**10.** Please ensure consistency in the terminology used for 'IWC-weighted $RH_{ice}$' and 'Ice MMR-weighted $RH_{ice}$' throughout the document.

**11. Line 386**
The phrase 'larger ICNC perturbations' could be misunderstood as referring to larger ice crystals. It is suggested to revise to 'larger perturbations in ICNC' for clarity.

**12. Reference**
The reference for 'Lee et al., 2021,' cited multiple times, is missing from the bibliography. Please add it.

**13. Lines 450 and 452**
The referenced information is actually found in Table 2, not Figures 3a and 6a. Please correct this.

**14. Line 468**
Should "larger" be "smaller"? Please check.

**15. Figure 8**
Correct the unit on the colorbar to "mg" instead of "m g"?

**16. Lines 518-519**
The punctuation at the end of each sentence could confuse readers about the sequence and relationship of these variables, potentially being mistaken for a dot product. Please consider revising for clarity.

**17. Lines 535-536**
It seems the described relationship might be reversed. Could the author please verify this?

**References:**

Kärcher, B., Burkhardt, U., Bier, A., Bock, L. and Ford, I.J., 2015. The microphysical pathway to contrail formation. *Journal of Geophysical Research: Atmospheres*, *120*(15), pp.7893-7927

Li, Y., Mahnke, C., Rohs, S., Bundke, U., Spelten, N., Dekoutsidis, G., Groß, S., Voigt, C., Schumann, U., Petzold, A. and Krämer, M., 2023. Upper-tropospheric slightly ice-subsaturated regions: frequency of occurrence and statistical evidence for the appearance of contrail cirrus. *Atmospheric chemistry and physics*, *23*(3), pp.2251-2271.

Marjani, S., Tesche, M., Bräuer, P., Sourdeval, O. and Quaas, J., 2022. Satellite observations of the impact of individual aircraft on ice crystal number in thin cirrus clouds. *Geophysical Research Letters*, *49*(5), p.e2021GL096173.

Unterstrasser, S., Gierens, K., SöLCH, I.N.G.O. and Lainer, M., 2017. Numerical simulations of homogeneously nucleated natural cirrus and contrail-cirrus. Part 1: How different are they?. *Meteorologische Zeitschrift*, *26*(6), pp.621-642.

Verma, P. and Burkhardt, U., 2022. Contrail formation within cirrus: ICON-LEM simulations of the impact of cirrus cloud properties on contrail formation. *Atmospheric Chemistry and Physics*, *22*(13), pp.8819-8842.

Voigt, C., Schumann, U., Minikin, A., Abdelmonem, A., Afchine, A., Borrmann, S., Boettcher, M., Buchholz, B., Bugliaro, L., Costa, A. and Curtius, J., 2017. ML-CIRRUS: The airborne experiment on natural cirrus and contrail cirrus with the high-altitude long-range research aircraft HALO. *Bulletin of the American Meteorological Society*, *98*(2), pp.271-288.